# Seroepidemiological and Molecular Survey for the Detection of SARS-CoV-2 Infection among Children in Iran, September 2020 to June 2021: 1-Year Cross-Sectional Study

**DOI:** 10.3390/microorganisms11071672

**Published:** 2023-06-27

**Authors:** Roxana Mansour Ghanaie, Idesbald Boone, Ahmad Reza Shamshiri, Abdollah Karimi, Arezu Amirali, Noushin Marhamati, Mohammad Hossein Rostami, Niloofar Pashaei, Shahriar Janbazi, Leila Azimi, Hannan Khodaei, Fatemeh Fallah, Tim Eckmanns, Andreas Jansen, Hamid Reza Baradaran, Maryam Momeny Ourimi, Saeed Maham, Ameneh Elikaei, Masoud Alebouyeh

**Affiliations:** 1Pediatric Infections Research Centre, Research Institute for Children’s Health, Shahid Beheshti University of Medical Sciences, Tehran 1546815514, Iran; ghanaieroxana@gmail.com (R.M.G.); dr.y.karimi@gmail.com (A.K.); arezuamirali24@gmail.com (A.A.); noushin.mt2@gmail.com (N.M.); leilaazimi1982@gmail.com (L.A.); hannankhodaei@yahoo.com (H.K.); dr_fallah@yahoo.com (F.F.); maham.sm@yahoo.com (S.M.); 2Department of Infectious Disease Epidemiology, Robert Koch Institute, 13353 Berlin, Germany; boonei@rki.de (I.B.); eckmannst@rki.de (T.E.); 3Research Center for Caries Prevention, Dentistry Research Institute, Tehran University of Medical Sciences, Tehran 1439955934, Iran; arshamshiri@tums.ac.ir; 4Department of Community Oral Health, School of Dentistry, Tehran University of Medical Sciences, Tehran 1439955934, Iran; 5Department of Microbiology, Faculty of Biological Sciences, Alzahra University, Tehran 1993893973, Iran; a.elikaei@alzahra.ac.ir; 6Snapp Group (Iran Internet Group), Tehran 1987913151, Iran; mhossein.rostami@gmail.com; 7Central Laboratory, Deputy of Public Health, Shahid Beheshti University of Medical Sciences, Tehran 1985717443, Iran; niloofarpashai@yahoo.com; 8Department of Health and Medical Medicine, Faculty of Medicine, Shahid Beheshti University of Medical Sciences, Tehran 1983969411, Iran; amir55amiri55@gmail.com; 9Centre for International Health Protection, Robert Koch Institute, 13353 Berlin, Germany; jansena@rki.de; 10Department of Epidemiology, School of Public Health, Iran University of Medical Sciences, Tehran 1449614535, Iran; baradaran98@gmail.com; 11Mofid Children’s Hospital, Shahid Beheshti University of Medical Sciences, Tehran 1546815514, Iran; www.maryam.momeny@yahoo.com

**Keywords:** SARS-CoV-2, COVID-19, children, seroprevalence, RT-qPCR, Iran

## Abstract

A population-based seroepidemiological and molecular survey on severe acute respiratory syndrome coronavirus 2 (SARS-CoV-2) was performed to detect induced antibodies to prior exposure and active infection of children aged 14 years or less in Tehran between 19 September 2020 and 21 June 2021. Moreover, correlations between the children’s demographic data and coronavirus disease 2019 (COVID-19) symptoms with the infection status were investigated. Out of 1517 participants, cardinal symptoms of COVID-19 (fever > 38 °C and/or cough and/or diarrhea) were detected in 18%, and serological history of SARS-CoV-2 infection and polymerase chain reaction (PCR) positivity were confirmed in 33.2% and 10.7% of the weighted population, respectively. The prevalence of SARS-CoV-2 infection was significantly higher among 10–14-year-old children. Active infection was significantly higher in symptomatic children and during autumn 2020 and spring 2021. The quantitative reverse transcription real-time PCR (RT-qPCR) positivity was significantly higher among families with a lower socioeconomic status, whereas no association between RT-qPCR or seropositivity was determined with household size, underlying diseases, or gender. In conclusion, high SARS-CoV-2 infection prevalence and seroprevalence were detected in children in Tehran in different seasons. Infection prevalence was significantly higher in older children and in those with a positive history of close contact with infected cases and/or lower socioeconomic status.

## 1. Introduction

Coronavirus disease 2019 (COVID-19) is an ongoing viral pandemic caused by severe acute respiratory syndrome coronavirus 2 (SARS-CoV-2). The main transmission route of this virus is inhalation of contaminated respiratory droplets; however, other modes of transmission for SARS-CoV-2 have also been described [1]. The disease is common as most patients are asymptomatic or may have common symptoms, such as fever, sore throat, rhinorrhea, cough, muscle aches, headache, shortness of breath, and gastrointestinal symptoms. The risk of hospitalization and mortality varies based on the virus variant, immune status, ethnicity, body mass index, gender, and underlying diseases [2,3].

The first confirmed cases of the novel coronavirus infection in children and neonates were reported in China in January 2020 [4]. Although some epidemiological studies showed that children are at lower risk of COVID-19 infection compared to adults [5,6], recent findings have demonstrated the existence of a similar viral load and, possibly, a similar secondary infection rate in children, suggesting that self-isolation and school closure may be the reason for the lower frequency of infections in this population [7,8,9]. Infected households can transmit SARS-CoV-2 to children in all age groups [10]; however, children are less affected by COVID-19, resulting in a small proportion of symptomatic infections that are characterized by generally milder symptoms [11]. The presence of untrained T lymphocyte cells against the virus and fewer Angiotensin converting enzyme 2 (ACE2) receptors in the nose of children were suggested as reasons for the lower frequency of symptoms in this population [12,13].

Although common symptoms of COVID-19 in children are fever and cough, other symptoms, including headache, nasal discharge, anosmia, and abdominal symptoms, were reported less frequently [14]. These symptoms are generally self-limited, and the patients recover, with few hospitalizations. The duration of COVID-19 disease between the onset and end of symptoms in children is shorter than in adults (6 days vs. 11–27.5 days, respectively) [15,16]. However, a longer duration of the disease could be predicted based on age, disease severity, a lower arterial oxygen partial pressure (PaO2 in mmHg) to fractional inspired oxygen (PaO_2_/FiO_2_) ratio, a weaker immune response, and the use of drugs such as steroids [15,16]. The risk factors for severe SARS-CoV-2 disease among children are not well known; however, multiple comorbidities, such as cardiovascular, neurological, and gastrointestinal conditions, may be involved [17]. In a study in Brazil, disease severity in children was also correlated with both social and economic status [18]. This diversity could be related to genetics, type of virus variant, immunity, and contact status.

With 7,563,728 reported cases, including 144,744 deaths (27 January 2023), Iran is the country most affected by COVID-19 among the Eastern Mediterranean region of the WHO, both in the number of cases and death rates [19]. However, local and global data about the incidence and prevalence of COVID-19 are scarce among children in comparison to other age groups [20,21,22]. In a serosurvey in 18 cities of Iran during the first COVID-19 wave in 2020, in which participants < 19 years of age constituted only 1.7% of the sample, a population-weighted seroprevalence of 15.4% was reported [21]. In a subsequent study conducted during the third COVID-19 wave in 16 cities of Iran, with only 5% of the participants being children ≥ 10–19 years of age, a higher population-weighted seroprevalence of 29.7% was reported [20]. At the time of this study, few data, especially from the Middle East, were available about the proportions of symptomatic and asymptomatic COVID-19 infections in children and their associations with demographics, seasonality, underlying diseases, the circulating virus variants, and the socioeconomic status. To understand these correlations, a community-based epidemiological survey on children, using both serological tests, which allow the detection of a history of prior infections with the virus, and molecular methods, which allow the detection of active infections with the virus, could provide valuable data. Due to the dynamics of the disease during the COVID-19 pandemic and limited data about the symptoms in infected children, this study investigated the seroprevalence and the quantitative reverse transcription real-time PCR (RT-qPCR) positivity of SARS-CoV-2 among a population of children in Tehran before the introduction of the COVID-19 vaccine, between 19 September 2020 and 21 June 2021. Moreover, associations of SARS-CoV-2 infection with demographic data and reported symptoms were analyzed in this population.

## 2. Materials and Methods

### 2.1. Study Design

To determine the seroprevalence of SARS-CoV-2 immunoglobulin G (IgG) antibodies and active SARS-CoV-2 infections in children of Tehran during autumn 2020, winter 2020–2021, and spring 2021, we used a cross-sectional design. To estimate sample size according to SARS-CoV-2 seroprevalence, calculations were performed with different scenarios, considering the prevalence of 5 to 50%, with a precision of 5 to 10% (95% confidence interval width). Accordingly, the number of 1574 participants was determined as the maximum required sample size. Due to concerns about the study participants attending public health centers and the expected low response rate during the pandemic, three different strategies were used to increase participation. For the first strategy, information on the study and a link for voluntary participation were provided to Tehran residents using “Snapp”, a ridesharing application with 2 million customers in Tehran (SNAPP strategy unit, Iran Internet Group). Upon registration, participants provided consent for participation in this study, as well as their age and gender. Clustering of the registered families was performed according to their recorded addresses and their distribution in the north, south, east, west, and central areas in 21 regions of Tehran. To increase the response rate, a phone call after initial registration, two reminder messages, a transportation discount, and a charge-free serological test for one of the parents were considered for the registered families. Sampling in public health centers, as the second strategy, was carried out according to available data from the Integrated Health System (SIB), the Iranian electronic health records system, to randomly select families who had children aged ≤ 14 years. Phone calls and reminder messages to selected families were used to increase the response rate. The third strategy was the voluntary sampling of children who were referred to the pediatric surgery and dentistry clinics in Mofid Children’s Hospital and healthy children who were referred to private laboratories for a general check-up. During the study period, COVID-19 vaccines had not yet been approved for use in children in Iran. Vaccination of children aged 12–18 years started in September 2021 [23].

### 2.2. Participants

Children residing in Tehran and aged 0 to 14 years at the time of recruitment were considered eligible to participate. Their parents or guardians signed a consent form before sampling. Families who refused to provide informed consent, or declared a contraindication to venipuncture, were excluded from the study. An orally administered questionnaire was used by physicians and trained staff to collect information from children, parents, or their guardians. The questions covered demographics (age, gender), household size, exposure to either suspected or confirmed cases of COVID-19, recent travelling, underlying diseases, history of positive SARS-CoV-2 tests, COVID-19 symptoms, and socioeconomic status. The existence of fever (>38 °C) and/or cough and/or diarrhea, defined as cardinal syndromes, in the presence or absence of other symptoms of COVID-19 in children (sore throat, fatigue, rhinorrhea, stomachache, headache, nausea/vomiting, wheeze, dyspnea, myalgia, chest pain, and change in smell or taste), defined as common COVID-19 symptoms, were considered as indicative for the statistical analysis [24]. The results of the SARS-CoV-2 tests and general clinical explanations were provided to all participants 24 h post-sampling.

### 2.3. Sampling

Blood and nasopharyngeal swab samples were obtained from each child and transported to the laboratory under cold chain conditions within 6 h. Blood samples were obtained by venipuncture and transferred into serum separator gel tubes. All swab samples were transported in viral transportation media. Serum samples were prepared after centrifugation and stored at −70 °C until use. The RNA was extracted from the swab samples using a High Pure Viral RNA extraction kit (F. Hoffmann-La Roche Ltd., Basel, Switzerland). The RNA extracts were stored at −70 °C until use for real-time PCR.

### 2.4. Serology Tests

Infected people with SARS-CoV-2 generally develop antibodies to SARS-CoV-2 antigens within 1–2 weeks of exposure, irrespective of their age [25]. To measure the history of infection with SARS-CoV-2, the IgG antibody against the S1 domain of the SARS-CoV-2 spike protein was measured using an anti-SARS-CoV-2 IgG antibody kit (Euroimmun, PerkinElmer Germany Diagnostics GmbH, Lübeck, Germany). The results were interpreted according to the manufacturer’s instructions.

### 2.5. Reverse Transcription Quantitative Real-Time Polymerase Chain Reaction (RT-qPCR)

Infection with SARS-CoV-2 during the sampling period was detected using a commercial kit (COVID-19 one-step RT-qPCR, Pishtaz Teb Diagnostics, Tehran, Iran). Two genomic loci of SARS-CoV-2, RdRp and N genes, and a human endogenous control gene for quality control were targeted to confirm the infection. The cycle threshold was calculated for each gene, and the results were interpreted according to the manufacturer’s instructions.

### 2.6. Statistical Analysis

For data clearance, we checked the “missing completely at random” (MCAR) assumption. The MCAR assumption was rejected according to the significant result of Little’s MCAR test. Therefore, assuming that the pattern of data missingness matched the “missing at random” (MAR), missing imputation was performed with the EM estimation. The software package IBM SPSS Statistics for Windows (version 25) (SPSS Inc., Chicago, IL, USA) was employed to carry out single imputations of variables with missing values.

Seroprevalence was estimated as the proportion of seropositive children with IgG SARS-CoV-2 antibodies. The prevalence of active SARS-CoV-2 infection was estimated as the proportion of children testing positive with RT-qPCR. For data weighting by age–gender–season (two categories for gender, three categories for age, and three categories for season), we used population data from the 2016 census, available from the Statistical Centre of Iran [26]. Frequency weights were calculated by dividing the size of the actual population in Tehran in each category by the number of participants in our study in the same categories. The survey data were analyzed using STATA version 14.2 (College Station, TX, USA). For descriptive data analysis, we report percentages with 95% confidence intervals (95% CI); however, because of intermittent epidemics over time, frequencies are reported by season. We used multivariable logistic regression analysis to evaluate associations between SARS-CoV-2 seropositivity and RT-qPCR positivity and demographics and potential risk factors. To adjust for confounding, we compared results from the univariable and multivariable analyses. A *p*-value less than 0.05 was considered significant.

### 2.7. Ethics Statements

The study was conducted in accordance with the Declaration of Helsinki and approved by the ethics committee of Shahid Beheshti University of Medical Sciences, Tehran, Iran (Code: IR.SBMU.RICH.REC.1399.050). Informed consent was confirmed by the Institutional Review Board (IRB).

## 3. Results

Overall, 1517 children throughout Tehran provided pairs of sera- and nasopharyngeal swab samples and answered the questionnaire during the study period (356 children in autumn 2020, 551 during winter 2020–2021, and 610 in spring 2021). In total, 1034 (68.2%) participants were boys, and 906 (60%) children were 4 years old or younger. Characteristics and related data about socioeconomic status and household size are shown in Table 1. Of the 728 participants (48% of study sample) who provided information about their geographic location, 11.8% originated from the northern, 33.2% from the southern, 34.6% from the eastern, 7.2% from the western, and 13.2% from the central areas of Tehran. Approximately 13.3% of children (202/1517) reported a history of contact with suspected or confirmed cases of COVID-19 (Table 1). Household members constituted 60% (121/202) of the contacts in children with a positive-contact history. According to the laboratory tests, 464 cases were ELISA-only positive, 101 cases were RT-qPCR-only positive, and 34 cases presented positive ELISA and RT-qPCR results. Seropositivity and RT-qPCR-positive test results accounted for 33.2% and 10.7% of the weighted population, respectively. The cardinal (fever > 38 °C and/or cough and/or diarrhea) and common COVID-19 symptoms (sore throat, fatigue, rhinorrhea, stomachache, headache, nausea/vomiting, wheeze, dyspnea, myalgia, chest pain, and change in smell or taste) were detected in 44.8% and 22.4% of the population-weighted RT-qPCR-confirmed cases of SARS-CoV-2, respectively, whereas asymptomatic infection was detected in 32.9% of them (Table 2). The seasonal distribution and population-weighted prevalence of RT-qPCR positivity and seropositivity for the studied samples are shown in Figure 1 and Appendix A.

The prevalence of SARS-CoV-2 IgG seropositivity was not associated with gender, season, socioeconomic status, and household size. However, SARS-CoV-2 IgG seropositivity was associated with older age (10–14 years age) as compared to younger children (0–4 years old) (aOR 1.45, 95% CI 1.02–2.06, *p*-value = 0.04) (Table 3). 

The prevalence of RT-qPCR positivity was higher in spring 2021 compared to winter 2020–2021 (OR 6.57, 95% CI 2.60–16.57, *p*-value < 0.001) and higher in symptomatic children compared to asymptomatic ones (OR 4.62 and 2.11 for the cardinal and common COVID-19 symptoms, respectively) (Table 4). According to the multivariable data analysis, RT-qPCR positivity was higher among children in the low socioeconomic group, and participants with smoking parents had lower odds of RT-qPCR positivity compared to those with nonsmoking parents (aOR 0.50, 95% CI 0.28–0.90, *p*-value = 0.02). RT-qPCR positivity was relatively higher in older age (10–14 years age) as compared to younger children (0–4 years old), although it was only on the borderline of statistical significance (aOR 1.72, 95% CI 0.99–3.01, *p*-value = 0.05) (Table 4).

## 4. Discussion

This study provides data on the seroprevalence and RT-qPCR positivity of SARS-CoV-2 infection and their association with demographics (age, sex), season, socioeconomic status, and COVID-19 symptoms among children ≤ 14 years of age. Our findings confirm that the infection can present either as asymptomatic or symptomatic in children, indicating that children can play a role as a silent reservoir of SARS-CoV-2 in the community. In a similar study in Brazil in 2020, approximately 15% of children were reported as asymptomatic [27]. Similarly, the pooled proportion of test-positive asymptomatic children from 13 different studies was estimated as 21.1% (95% CI: 14.0–28.1%) in a review of the literature [28].

In the present study, a relative increase in seroprevalence of SARS-CoV-2 was observed in children in the higher age groups. In a study conducted in 2021 in Italy, the seroprevalence in the age groups younger than 5 years, 6–11, and 12–17 years was 18%, 37.6%, and 43.7%, respectively [29]. In the United States, a study conducted between April and May 2020 reported a seroprevalence of 0.8% and 4.5% in children > 4 and 5–14 years of age, respectively [30]. A study in Switzerland reported a higher seroprevalence in 10–19-year-old children (9.6%) compared with the 5–9-year-old group (0.8%) [31].

In the present study, there was no difference in seropositivity or RT-qPCR positivity between girls and boys. A slightly higher frequency of infection (RT-qPCR positivity) among boys was reported in a study in China (19.5% in boys compared to 14.5% in girls) [32]. In contrast, a higher frequency of infection among girls was reported in a study in the United States (11.8% vs. 10.1% in boys) [33].

In this study, the rate of SARS-CoV-2 infection estimated using serological and molecular methods in children was not significantly associated with a larger household size. A study in Brazil in 2021 reported a higher prevalence of infection in families with more than three members (29.8%) compared to fewer members (23%) [27]. A higher secondary attack rate was reported in a study in Norway among families with more than six family members (25% from parents to children). The study showed a higher secondary transmission in larger families, with more contact with family members and more cramped living conditions favoring infection in children [34]. In a study in Iran, 31% of affected children had a history of contact with a suspected or infected family member [35]. During the study period in Tehran, distance learning was implemented, and our findings suggest that family members played a prominent role to the household transmission of SARS-CoV-2 infection to children. Higher exposure to COVID-19 patients appears to be the main risk factor for the secondary transmission of SARS-CoV-2 to the family members in this population. The measured higher RT-qPCR positivity among children in the low socioeconomic group in our study could be explained based on the limited access to hand sanitizers and personal protective equipment, such as masks, living in a more densely populated area, and the different housing conditions, making isolation of the infected household members difficult [36,37]. In the present study, a higher prevalence of SARS-CoV-2 infection was detected according to the PCR results in spring of 2021, which is consistent with the early course of a new wave of COVID-19 (April to June 2021) caused by more contagious variants (i.e., alpha and delta variants) in Iran [38,39]. Our findings showed that SARS-CoV-2 infection was significantly lower in children with smoking parents compared with the nonsmoking ones. Although the serological results did not confirm such a correlation and the evidence and quality of this association are still unclear, the negative association between smoking and SARS-CoV-2 positivity was similarly reported in other studies [40].

This study has several limitations. We suspect that children with some clinical symptoms may have been more likely to accept our invitation to participate in the study; accordingly, selection bias may have been introduced in different settings. Dissatisfaction and unawareness of the parents resulted in missing data for some questions. Willingness to participate in this study was low due to fear of attending hospitals/public health centers, incorrect opinions about the minor impact of COVID-19 on children during the primary course of sampling, fear of blood sampling in children despite offering incentives to the parents such as involving free-of-charge serological tests and a transportation discount. The greater rate of participation of symptomatic families during the peaks, issues related to self-reporting for socioeconomic status, and some other variables, such as symptoms, are among other limitations in this study. Moreover, we were not able to estimate the duration of COVID-19 symptoms in children to compare viral shedding in the symptomatic and asymptomatic infected children due to the families’ unwillingness to allow the children to participate in a follow-up study and revisit the hospital or public health centers.

## 5. Conclusions

The special feature of this SARS-CoV-2 study is its large sample size in a specific population, namely, children up to 14 years of age in Tehran, which had previously been studied only to a limited extent. During the study period, high seroprevalence, due to prior exposure, and high prevalence of active infection with SARS-CoV-2 were detected in children. Children in the higher age group who had been in contact with suspected or confirmed cases within the household and those living in families with a lower socioeconomic status were more likely to be infected with SARS-CoV-2. The SARS-CoV-2 infection occurred in children in the presence or absence of cardinal or common COVID-19 symptoms. Although seasonality was confirmed based on the RT-qPCR positivity results, no significant change in the seropositivity rate was detected in different seasons. Further studies are needed to improve our understanding of the impacts of vaccines on reinfection, changes in COVID-19 symptoms, and post-COVID-19 sequelae in children.

## Figures and Tables

**Figure 1 microorganisms-11-01672-f001:**
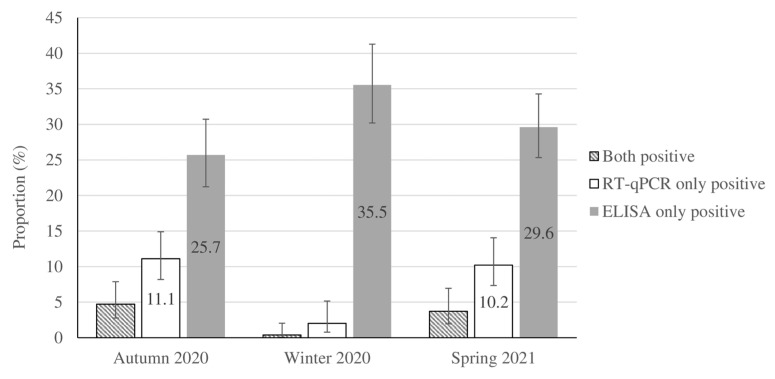
Adjusted proportion of positive quantitative reverse transcription real-time PCR (RT-qPCR) and ELISA test results for COVID-19 in a population of children in Tehran during autumn 2020 to spring 2021. The results are presented according to weighting considering age, gender, and season. Bars indicate 95% confidence intervals.

**Table 1 microorganisms-11-01672-t001:** Basic characteristics of the children participating in the study in Tehran from autumn 2020 to spring 2021 (*n* = 1517) *.

Variables	Categories	*n* (%)
Age groups (years old)	0–4	906 (59.72)
	5–9	417 (27.49)
	10–14	194 (12.79)
Gender	Female	483 (31.84)
	Male	1034 (68.16)
Season	Autumn 2020	356 (23.47)
	Winter 2020	551 (36.32)
	Spring 2021	610 (40.21)
Smoking (parents)	No	1248 (82.27)
	Yes	269 (17.73)
Socioeconomic status	Medium and high	867 (57.15)
	Low	650 (42.85)
Household size	2–3	600 (39.55
	4	658 (43.38)
	>4	259 (17.07)
Symptomatic	Asymptomatic	975 (64.27)
	Common symptoms ^1^	269 (17.70)
	Cardinal Symptoms ^2^	273 (18)
Contact with suspected or confirmed cases during 4 last weeks	Yes	202 (13.3)
	No	1206 (79.5)
	Unknown	109 (7.2)

* The table was generated based on imputed data. ^1^ Common symptoms: at least one of the followings symptoms reported: sore throat, fatigue, rhinorrhea, stomachache, headache, nausea/vomiting, wheeze, dyspnea, myalgia, chest pain, and change in smell or taste. ^2^ Cardinal symptoms: fever (>38 °C) and/or cough and/or diarrhea.

**Table 2 microorganisms-11-01672-t002:** Frequency of COVID-19 self-reported symptoms among children by RT-qPCR positivity for SARS-CoV-2, Tehran, autumn 2020 to spring 2021 (*n* = 1517).

Symptoms *	RT-qPCR Positive (*n* = 135)	%	RT-qPCR Negative (*n* = 1382)	%
Sore throat	33	24.4	92	6.7
Fatigue	32	23.7	85	6.2
Fever > 38 °C	31	22.9	117	8.5
Cough	30	22.2	100	7.2
Rhinorrhea	20	14.8	103	7.4
Chills	19	14	47	3.4
Stomach ache	18	13.3	120	8.7
Headache	17	12.6	66	4.7
Nausea/vomiting	16	11.8	65	4.7
Diarrhea	14	10.3	63	4.5
Wheeze	10	6.6	31	2.2
Dyspnea	8	5.9	14	1
Myalgia	7	5.2	17	1.2
Chest pain	4	2.9	15	1.1
Other respiratory	3	2.2	7	0.5
Change in smell and/or taste	1	0.7	8	0.57

* All symptoms, except chills, were recorded with ≥10% missing data in the questionnaire. The exact number of missing values can be found in Appendix A (Supplementary Material). RT-qPCR—quantitative reverse transcription real-time PCR.

**Table 3 microorganisms-11-01672-t003:** Associations of SARS-CoV-2 IgG seropositivity (ELISA) with demographic and potential risk factors among children in Tehran (autumn 2020 to spring 2021).

	Bivariate Data Analysis			Multivariable Data Analysis		
	Unadjusted OR	95% CI	*p*-Value	Adjusted OR	95% CI	*p*-Value
Age						
0–4	Reference			Reference		
5–9	1.29	0.99–1.68	**0.05**	1.23	0.93–1.62	0.13
10–14	1.53	1.08–2.17	**0.01**	1.45	1.02–2.06	**0.04**
Gender						
Male	Reference			Reference		
Female	0.92	0.70–1.20	0.54	0.91	0.69–1.2	0.53
Season						
Winter 2020–2021	Reference			Reference		
Autumn 2020	0.77	0.55–1.09	0.14	0.73	0.51–1.05	0.09
Spring 2021	0.89	0.64–1.23	0.48	0.88	0.64–1.22	0.47
Socioeconomic status						
Low	Reference			Reference		
Medium and high	0.87	0.66–1.15	0.34	0.92	0.69–1.23	0.59
Family members						
2–3	Reference			Reference		
4	1.24	0.92–1.66	0.14	1.12	0.83–1.51	0.43
>4	1.32	0.89–1.97	0.15	0.17	0.78–1.75	0.44
Smoker (parents)						
No	Reference			Reference		
Yes	1.08	0.76–1.53	0.64	1.05	0.73–1.5	0.77
Contact with confirmed/suspected cases during last 4 weeks						
No	Reference			Reference		
Yes	1.15	0.81–1.65	0.43	1.37	0.91–2.07	0.13
Unknown	1.14	0.69–1.88	0.61	1.23	0.75–2.04	0.41
Symptomatic						
Asymptomatic ^1^	Reference			Reference		
Common symptoms ^2^	1.29	0.91–1.82	0.14	1.20	0.84–1.73	0.30
Cardinal symptoms ^3^	0.84	0.59–1.19	0.33	0.80	0.53–1.19	0.27

^1^ Absence of common and cardinal COVID-19 symptoms in children. ^2^ At least one of the following symptoms: sore throat, fatigue, rhinorrhea, stomachache, headache, nausea/vomiting, wheeze, dyspnea, myalgia, chest pain, and change in smell or taste. ^3^ Cardinal symptoms in children, fever (>38 °C) and/or cough and/or diarrhea. Numbers in bold represent *p*-values that are statistically significant.

**Table 4 microorganisms-11-01672-t004:** Association of SARS-CoV-2 RT-qPCR positivity and demographic and potential risk factors with positive results among children in Tehran (autumn 2020 to spring 2021).

	Bivariate Data Analysis			Multivariable Data Analysis		
	Unadjusted OR	95% CI	*p*-Value	Adjusted OR	95% CI	*p*-Value
Age						
0–4	Reference			Reference		
5–9	1.29	0.84–1.99	0.23	1.20	0.74–1.95	0.45
10–14	1.86	1.14–3.05	**0.01**	1.72	0.99–3.01	0.05
Gender						
Male	Reference			Reference		
Female	1.12	0.74–1.70	0.56	0.98	0.62–1.55	0.95
Season						
Winter 2020–2021	Reference			Reference		
Autumn 2020	7.63	3.05–19.07	**<0.001**	4.60	1.67–12.62	**0.003**
Spring 2021	6.57	2.60–16.57	**<0.001**	6.10	2.38–15.64	**<0.001**
Socioeconomic status						
Low	Reference			Reference		
Medium and high	1.01	0.65–1.55	0.95	0.42	0.25–0.69	**0.001**
Household size						
2–3	Reference			Reference		
4	1.26	0.79–2.01	0.32	1.20	0.71–2.03	0.48
>4	0.86	0.46–1.62	0.65	0.92	0.47–1.80	0.81
Smoking (parents)						
No	Reference			Reference		
Yes	0.62	0.36–1.08	0.095	0.50	0.28–0.90	**0.02**
Contact with confirmed/suspected cases during last 4 weeks						
No	Reference			Reference		
Yes	5.84	3.62–9.42	**<0.001**	3.76	1.99–7.14	**<0.001**
Unknown	4.13	2.20–7.74	**<0.001**	3.53	1.83–6.79	**<0.001**
Symptomatic						
Asymptomatic ^1^	Reference			Reference		
Common symptoms ^2^	2.11	1.16–3.83	**0.013**	1.32	0.69–2.53	0.40
Cardinal symptoms ^3^	4.62	2.85–7.49	**<0.001**	3.07	1.73–5.44	**<0.001**

^1^ Absence of common and cardinal COVID-19 symptoms in children. ^2^ Common COVID-19 symptoms in children: sore throat, fatigue, rhinorrhea, stomachache, headache, nausea/vomiting, wheeze, dyspnea, myalgia, chest pain, and change in smell or taste. ^3^ Cardinal symptoms in children, fever (>38 °C) and/or cough and/or diarrhea. Numbers in bold face represent *p*-values that are statistically significant.

## Data Availability

The data presented in this study are available upon reasonable request from the corresponding author.

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
