# Peer review of "Seroepidemiological and Molecular Survey for the Detection of SARS-CoV-2 Infection among Children in Iran, September 2020 to June 2021: 1-Year Cross-Sectional Study"

_microorganisms, 2023, doi:10.3390/microorganisms11071672_

Round 1

Reviewer 1 Report

Dear Authors

I think the aim of the study must be better explained, also in the abstract. The data are not new in literature thus I think that you have to let emerge and underline some relevant and new information from your data.

For example discuss how long was covid infection (if you have this data) between children (see and cite doi:10.3390/microorganisms8091327).

Moreover discuss, If possible, the outcome of the infected children and create a table/figure representing the most principle differences in symptoms between different age groups.

Revise English all over the text.

Moderate revision needed.

Author Response

  1. I think the aim of the study must be better explained, also in the abstract.
  2. The data are not new in literature thus I think that you have to let emerge and underline some relevant and new information from your data.For example discuss how long was covid infection (if you have this data) between children (see and cite doi:10.3390/microorganisms8091327).
  3. Moreover discuss, If possible, the outcome of the infected children and create a table/figure representing the most principle differences in symptoms between different age groups.
  4. Revise English all over the text."Please see the attachment."

Reviewer 2 Report

The manuscript  need to be improve in some important parts.  As major points: the introduction  should be shortened; the discussion lacks a contextualization regarding the vaccination policy and uptake and type of vaccine in use during the observation period together with some  important details regarding the strains circulating at that time in the country . The conclusion is not very clear especially in the last part and in consideration of the main aims of the work. The socioeconomic status,i.e,  as point associated with a higher infection rate among children, need to be better explained. Does it mean the impossibility to maintain the isolation among infected households, for example?  The cited further studies  to control the spread of the diseases in children in an era (2023)  of a low circulation of  virus variants less capable of defining a more severe clinical picture and a possible low number of cases,  how could they  be carried  out in the country context?

 Minor points:

 Abstract: line 25  earlier and recent , what does it mean?

Introduction; line 51 studied in children are  limited , not really true, please rephrase.

MM: serology test to evaluated previous infection  must consider N target and non only Spike protein. 

Results: line 200 what about the others ?; line 213, did the authors test a subsample  with an additional kit to have a confirmation of the results? did they use references in the tests? 

Discussion: line 264, please explain more about  the relative increase  in the older age group, is there any statistical support for this point? 

Edite  the english through the  text. 

Author Response

Dear Editor-in-chief, dear reviewers,

Thank you for your response and your valuable comments and suggestions on the revision of the manuscript “Seroepidemiological and molecular survey for detection of SARS-CoV-2 infection among children in Iran, September 2020 to June 2021: one-year cross-sectional study.”

The manuscript has been professionally proofread (see track changes). Below, we addressed the reviewers' questions and described any changes made to the manuscript.

We sincerely hope that the revised manuscript satisfactorily addresses the issues raised.

Kind regards,

Masoud Alebouyeh on behalf of the coauthors

Comments of reviewer 2:

  1. The introduction should be shortened

Response: Thank you for your remark. We have revised and shortened the introduction in lines 78 ff. [Ref. 12], 82 ff. [Ref. 15], and 94 ff.

  1. The discussion lacks a contextualization regarding the vaccination policy and uptake and type of vaccine in use during the observation period together with some important details regarding the strains circulating at that time in the country.

Response: Thank you for your remark, as described in lines 143-145, COVID-19 vaccines had not yet been approved for use in children in Iran during the study period. Vaccination of children aged 12–18 years started in September 2021 in Iran. Regarding the reviewer’s comment, the correlation of the observed increase in the frequency of the PCR positive cases and a new wave of alpha and delta variants, which were more contagious, and a suggestion about accelerating the introduction of the COVID-19 vaccine in children before the new wave was added to the discussion section.

In the case of circulating strains at the time of the study period, the clades that circulated in Iran in the third wave from October to December 2020 at the beginning of our study were GH and GR. In the fourth wave from the beginning of April to June 2021 at the end of the study, GRY (alpha variant) and GK (delta variant) were reported as dominant clades (doi: 10.1111/irv.13135.).

The following sentence in the revised manuscript described this link (lines 338-342):

“According to the PCR results, a higher prevalence of SARS-CoV-2 infections was detected in the spring of 2021, which is consistent with the early course of a new wave of COVID-19 (April to June 2021) caused by more contagious variants (i.e., alpha and delta variants) in Iran [40, 41].”

  1. The conclusion is not very clear, especially in the last part and in consideration of the main aims of the work.

Response: Thank you for this remark. We have described the aims of the study more detail and more clearly in the revised manuscript (Lines 26-30 of the abstract and lines 104-116 of the Introduction. The conclusion was also revised and presented based on the revised aims and objectives (Lines 359-371).

  1. The socioeconomic status, i.e, as point associated with a higher infection rate among children, need to be better explained. Does it mean the impossibility to maintain the isolation among infected households, for example? 

Response: Thank you for this question, a likely explanation for the association between RT-qPCR positivity and a lower socioeconomic status (SES) was included to the manuscript (Lines 327-332) as shown below:

“The measured higher RT-qPCR positivity among children in the low socioeconomic group in our study could be explained based on the limited access to hand sanitisers and personal protective equipment, such as masks, living in a more densely populated area and the different housing conditions, making isolation of the infected household members difficult [38, 39].”

  1. The cited further studies to control the spread of the diseases in children in an era (2023)  of a low circulation of  virus variants less capable of defining a more severe clinical picture and a possible low number of cases, how could they  be carried  out in the country context?

Response: The sentence was rephrased as follows in the conclusions (lines 369-371):

Further studies are needed to improve our understanding of the impact of vaccines on reinfection, changes in COVID-19 symptoms, and post-COVID-19 sequelae in children.

  1. Abstract: line 25 earlier and recent, what does it mean?

Response: These words refer to the previous and recent active SARS-CoV-2 infection. The sentence was revised as shown below:

“A population-based seroepidemiological and molecular survey was performed to detect induced antibodies to prior exposure in sera and active SARS-CoV-2 infection from nasopharyngeal swab samples of children aged 14 years or less in Tehran between 19 September 2020 and 21 June 2021.”

  1. Introduction; line 51 studied in children are limited, not really true, please rephrase.

Response: Thank you, we agree that currently this sentence is not true anymore. We have deleted the sentence.

  1. MM: serology test to evaluated previous infection must consider N target and non-only Spike protein.

Response: This study was done from September 2020 to June 2021; during this period SARS-CoV-2 vaccines were not approved to children in Iran. Detection of anti-N in parallel to anti-Spike proteins could help to differentiate between vaccine-based induced antibody and natural infections. However, this was not the necessary since SARS-CoV-2 vaccination was not yet approved for childred during the study period in Iran.

  1. Line 200 what about the others?

Response: “Approximately 13.3% of children (202/1517) reported a history of contact with suspected or confirmed cases of COVID-19.” Details about the others, who reported “no contact” or are recorded as “unknown” are presented in Table 1. Table 1 was referred to in the revised manuscript.

  1. Line 213, did the authors test a subsample with an additional kit to have a confirmation of the results? Did they use references in the tests?

Response: Yes, we used the Wantai kit to confirm the results. The Wantai ELISA kit is based on the receptor binding domain of the SARS-CoV-2 spike protein. As described in lines 237-239, the results of the Euroimmun kit were confirmed by the Wantai kit with a positive predictive value of 91.2% (187/205).

  1. Line 264, please explain more about the relative increase in the older age group, is there any statistical support for this point? 

Response: Thank you for your remark. We have rephrased the sentence about RT-qPCR positivity and its association with age groups. As presented in Tables 3 and 4, children in the age group 10-14 y showed a higher OR compared to other age groups, which was significant statistically. P values were added to the results. Related sentences are shown below:

[SARS-CoV-2 IgG seropositivity was associated with older age (10–14 years age) as compared to younger children (0–4 years old) (OR 1.53, 95% CI 1.08–2.17, p value= 0.015) (Table 3).]

[Similar to SARS-CoV-2 IgG seropositivity results, RT-qPCR positivity was associated with older age (10–14 years age) as compared to younger children (0–4 years old) (OR 1.86, 95% CI 1.14–3.05, p value= 0.01) (Table 4).]

  1. Edit the English through the text.

Response: The manuscript has been professionally proofread (see track changes).
